# Template Transformer Networks for Image Segmentation

**Matthew Chung Hai Lee**[1,2]                                    MLEE@HEARTFLOW.COM
**Kersten Petersen**[1]                                          KPETERSEN@HEARTFLOW.COM
**Nick Pawlowski**[2]                                          N.PAWLOWSKI16@IMPERIAL.AC.UK
**Ben Glocker**[*1,2]                                          B.GLOCKER@IMPERIAL.AC.UK
**Michiel Schaap**[*1,2]                                          MSCHAAP@HEARTFLOW.COM

[1] *HeartFlow, California, CA 94063, USA*

[2] *Biomedical Image Analysis Group, Imperial College London, London SW7 2AZ, UK*

**Editors:** Under Review for MIDL 2019

## Abstract

In this paper we introduce and compare different approaches for incorporating shape prior information into neural network based image segmentation. Specifically, we introduce the concept of template transformer networks where a shape template is deformed to match the underlying structure of interest through an end-to-end trained spatial transformer network. This has the advantage of explicitly enforcing shape priors and is free of discretisation artefacts by providing a soft partial volume segmentation. We also introduce a simple yet effective way of incorporating priors in state-of-the-art pixel-wise binary classification methods such as fully convolutional networks and U-net. Here, the template shape is given as an additional input channel, incorporating this information significantly reduces false positives. We report results on sub-voxel segmentation of coronary lumen structures in cardiac computed tomography showing the benefit of incorporating priors in neural network based image segmentation.

**Keywords:** Image Segmentation, Shape Priors, Neural Networks, Template Deformation, Image Registration

## 1. Introduction

Segmentation of anatomical structures can be greatly improved by incorporating priors on shape, assuming population wide regularities are observed, or that expert knowledge is available. Shape priors help to reduce the search space of potential solutions for machine learning algorithms, improving the accuracy and plausibility of solutions (Nosrati and Hamarneh, 2016). Priors are particularly useful when data is ambiguous, corrupt, exhibits a low signal-to-noise ratio or if training data is scarce.

## 2. Method

Traditional template deformation models require the definition of an image-to-segmentation matching function as an approximation or surrogate to the actual segmentation objective. Iterative optimisation is then used to incrementally update the transformation parameters in order to maximize agreement between a template and the image to be segmented. In

---

* Shared senior authorship.

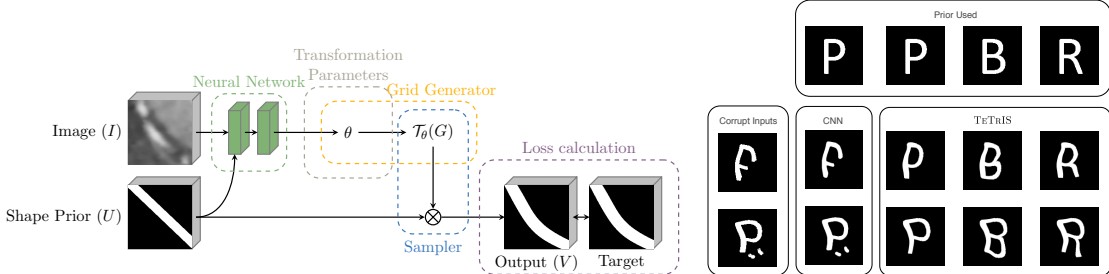

Figure 1: On the left, a diagram showing how TETRIS takes as input an image and a shape prior in the form of a partial volume image and produces a set of parameters for a transformation. This transformation is applied to the prior and the loss is calculated between the deformed prior and the target segmentation. On the right, an illustrative example of where a deformation model can extrapolate well outside of the distribution of the training data compared to a standard convolutional neural network.

contrast, our method makes use of neural network based registration, which only requires the computation of a corresponding loss function (equivalent to the matching function) during *training* time. This important difference means we no longer need to approximate our actual segmentation function via an intensity-based surrogate and can directly optimise for the task at hand.

By combining template deformation with neural networks, we mitigate the key problem with traditional template deformation models, that being the need to hand craft a good image to segmentation alignment function. The source of this problem, as with any registration techniques, lies in the fact that a loss calculation must be made during test time to update the deformation field parameters $\theta$. By utilising Spatial Transformer Networks (Jaderberg et al., 2015) to produce $\theta$ during test time and instead updating a neural network $f_\psi$ during training, we can train a registration model with the true segmentation loss function (based on alignment between prior and reference segmentation) avoiding the need for surrogate functions at test time. We provide a schematic of our model in Fig. 1 and provide full details in (Lee et al., 2019).

Due to the ill-posed nature of registration problems, it is common to constrain deformation fields by adding a regularisation term to the optimisation problem that favours some desired property, such as locally smooth deformations, or an $L_2$ penalty on the vector field itself to favour minimum displacement solutions.

We investigate two regularisation terms: $\mathcal{L}_{L_2}$ which penalises the the $L_2$-norm of the field and $\mathcal{L}_{\text{smooth}}$ which penalises the sum of squared second order derivatives.

## 3. Experiments

We use two baseline models to compare the three models we present i) the residual fully convolutional network (FCN) and ii) a residual U-net architecture utilising the implementations from (Pawlowski et al., 2017) using residual blocks from (He et al., 2015). Full training and architecture details can be found in (Lee et al., 2019).

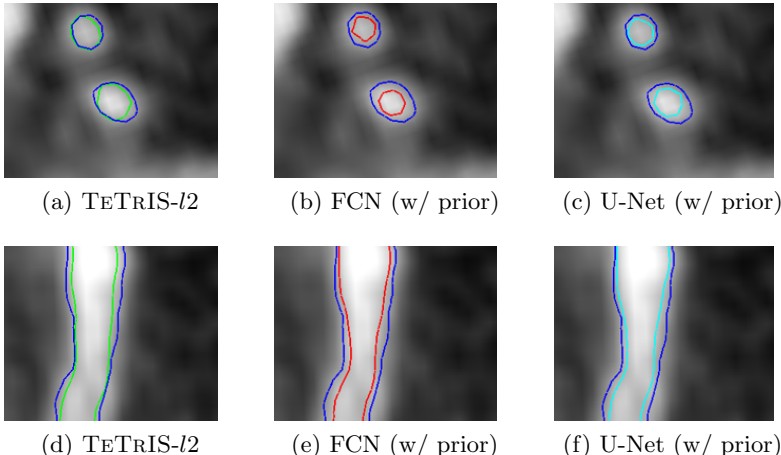



(a) TeTrIS-*l2*  (b) FCN (w/ prior)  (c) U-Net (w/ prior)

(d) TeTrIS-*l2*  (e) FCN (w/ prior)  (f) U-Net (w/ prior)



Figure 2: Qualitative results shown as contours for the different methods where the blue, green, red and cyan contours are of the target segmentation, TeTrIS, FCN (with prior) and U-Net (with prior) respectively. Rows show orthogonal views of the left anterior descending artery, near the first diagonal bifurcation where TeTrIS outperforms other methods

We also present results on naively incorporating shape priors into these state-of-the-art models. We do this by feeding the networks two channels of input, the image to be segmented and the prior that we have of the image at that location. This alternative method is a very simple extension of existing state-of-the-art approaches, computationally cheap and easy to implement.

We train our network on a set of 274 annotated cardiac CT volumes with 0.5 millimetre isotropic spacing and reserving 138 volumes for validation and an additional 136 for testing. The ground truth labels obtained through manual expert segmentation are in the form of partial volumes. Shape priors are created by tubing along a centerline generated from a semi automatic method with fixed radius of 1 mm which results in a partial volume prior.

Result show our model strikes a better balance between performance and model complexity, even though our model is restricted in the sense that it can only perform deformations of a prior, we argue this can be an advantage where shape guarantees are important.

Table 1: Quantitative Segmentation Results on Test Cases

|  | Cross Entropy | Connected Components | Dice Score | Hausdorff Distance | Trainable Parameters |
|---|---|---|---|---|---|
| U-Net | 0.01219 | 26.4 | 0.336 | 73.41 | 11.94M |
| FCN | 0.01190 | 27.0 | 0.406 | 59.94 | 13.74M |
| U-Net (w/ prior) | 0.00186 | 1.0 | 0.854 | 2.86 | 11.95M |
| FCN (w/ prior) | 0.00163 | 1.1 | 0.790 | 2.87 | 13.75M |
| TeTrIS-no-reg | 0.00162 | 1.0 | 0.779 | 3.20 | 1.38M |
| TeTrIS-*l2* | 0.00160 | 1.0 | 0.787 | 3.36 | 1.38M |
| TeTrIS-smooth | 0.00163 | 1.0 | 0.768 | 3.55 | 1.38M |

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
