# OpenReview forum: "Template Transformer Networks for Image Segmentation"
_MIDL.io/2019/Conference/Abstract — MIDL Abstract 2019_

### Official Review · AnonReviewer1 · 2019-04-24
**Idea of adding shape is interesting**

**Rating:** 3
**Confidence:** 3

**Review:**

The quality is very good. and the clarity is very good.
The abstract is well written and easy to follow.
The validation is solid.
The significance of the work is to introduce the shape prior as an input.
It will guide the network to understand or capture the underline shape constraint better.
The limitation is that the shape priors are created by a semi-automatic method. It would be better to further automate that part.

---

### Official Review · AnonReviewer2 · 2019-04-30
**Interesting work on incorporating shape priors into neural networks**

**Rating:** 4
**Confidence:** 3

**Review:**

This paper presents a method for incorporating shape prior into  neural networks to improve segmentation.
This is achieved by introducing template  transformer network where the shape is transformed to match the target using spatial transformer networks. In addition, a method to incorporate priors into standard CNNs is described.
The work is well motivated. The method is very briefly described without details (obvious due to space limitations).
Evaluation is performed using coronary artery segmentation and the presented method is compared with standard convolutional neural networks.
The writing is very condense, so it is hardly possible to evaluate the details, especially on methodology, but very interesting and novel methodology is described, with excellent results.

---

### Decision · Program_Chairs · 2019-05-06
**Acceptance Decision**

Accept